# The Lacewings *Micromus angulatus* and *Chrysoperla carnea* as Predators of the Rhododendron Aphid, *Illinoia lambersi*, Under Different Temperature Regimes

**DOI:** 10.3390/insects17010046

**Published:** 2025-12-30

**Authors:** Marie Froyen, Robin Beckx, Ellen Peeters, Wan-Yi Liao, Joachim Audenaert, Ruth Verhoeven, Alberto Pozzebon, Bruno Gobin, Patrick De Clercq

**Affiliations:** 1Department of Plants and Crops, Ghent University, Coupure Links 653, B-9000 Ghent, Belgiumrobin.beckx@ugent.be (R.B.); ellen.sara.peeters@hotmail.com (E.P.); wyliao535@gmail.com (W.-Y.L.); 2Department of Agronomy, Food, Natural Resources, Animals and Environment, University of Padua, Viale dell’Università 16, I-35020 Legnaro, Italy; alberto.pozzebon@unipd.it; 3Viaverda, Schaessestraat 18, B-9070 Destelbergen, Belgium; joachim.audenaert@viaverda.be (J.A.); ruth.verhoeven@viaverda.be (R.V.); bruno.gobin@viaverda.be (B.G.)

**Keywords:** predators, Chrysopidae, Hemerobiidae, Aphididae, rhododendron pests, ornamental crops, biological control

## Abstract

The aphid *Illinoia lambersi* is a key pest of *Rhododendron* in Northwestern Europe, causing leaf deformation, reduced flowering, and overall loss of plant quality. To support the development of pesticide-reducing strategies, we evaluated the potential of the brown lacewing *Micromus angulatus* as a biological control agent against this aphid. In laboratory tests, we quantified the predation capacity of *M. angulatus* larvae and adults at temperatures typical of spring and summer growing conditions (15, 20 and 25 °C), and compared their performance with that of larvae of the widely used green lacewing *Chrysoperla carnea*. Brown lacewing larvae of the third instar, as well as male and female adults of *M. angulatus*, consistently reduced aphid numbers across all tested temperatures. At 15 °C, its third-instar larvae performed similarly to those of *C. carnea*, whereas at 20 °C the green lacewing showed higher predation rates. Predation by female adults of the brown lacewing was similar across all tested temperatures. Female adults of this species were more effective than males. Because *M. angulatus* is predatory throughout both larval and adult stages, it may provide sustained aphid suppression over longer periods than biological control agents that feed only as larvae, like *C. carnea*. Our findings highlight the potential of *M. angulatus* as a biological control agent against *I. lambersi*, particularly under the cool spring conditions at the start of the rhododendron growing season. Further field validation is needed to confirm laboratory results and assess performance across different rhododendron cultivars.

## 1. Introduction

Belgium is the largest producer of pot azaleas (*Rhododendron simsii* hybrids) in Europe, with more than 80% of European production concentrated in the Flanders region. In addition to pot azaleas cultivated as houseplants, many other *Rhododendron* species and varieties are grown as garden plants. In Flanders, *Rhododendron* spp. are cultivated both outdoors and in greenhouses [1]. In both settings, rhododendron plants are susceptible to a wide range of pests and diseases [2], which to date have been managed primarily with chemical pesticides. Based on an unpublished questionnaire by research center Viaverda, Flemish growers apply an average of seven chemical treatments per year on rhododendron crops.

The main aphid pest on rhododendron plants is *Illionia* (*Masonaphis*) *lambersi* (MacGillivray). The aphid is native to North America but was introduced into Europe in the early 1970s and is now widespread on the continent [3,4,5]. It is also present in parts of South America and Asia [6]. This spindle-shaped aphid mainly lives on *Rhododendron* spp., but can also be found on other plants like *Kalmia latifolia* and *Ilex aquifolium.* On *Rhododendron*, the aphid infests shoots, leaves, and flower buds. The damage pattern differs between evergreen and deciduous rhododendrons, but usually involves malformation of the leaves, increased leaf fall, reduced flower production, and, during heavy infestations, a withered appearance of the plant. Additionally, the aphid’s exuviae reduce the aesthetic value of the plants [4,5,7].

To reduce pesticide use in rhododendron production, it is essential to develop IPM strategies that ideally incorporate the conservation or augmentative release of biological control agents. However, knowledge about the natural enemies of *I. lambersi* remains limited. Hille Ris Lambers [4] reported that effective attacks on the rhododendron aphid by coccinellid, syrphid, and chrysopid predators, as well as hymenopteran parasitoids, were scarce. In evergreen rhododendron varieties, the latter author attributed the lack of effective predation or parasitization to the fact that these arthropod natural enemies struggled to cope with the glandular trichomes, whereas the aphid was unaffected.

Among the range of predators and parasitoids that are commercially available to manage aphid outbreaks, lacewings (Neuroptera) have long been considered key aphid predators in various horticultural and agricultural cropping systems [8,9]. Although a considerable body of literature exists on the ecology of green lacewings of the Chrysopidae family and their use in biological control, far less is known about the biological control potential of brown lacewings belonging to the Hemerobiidae [9,10,11]. *Micromus angulatus* (Stephens) is a species of the family Hemerobiidae that is distributed widely in Europe, and in parts of North America and Eastern Asia [12]. Both the larvae and adults of *M. angulatus* feed on a variety of aphids and have been proposed for augmentative releases against aphid pests in greenhouse and field crops [9,11,13,14,15,16,17,18].

In this study, we evaluated the predation capacity of *M. angulatus* on *I. lambersi* under laboratory conditions at different temperature regimes, representing those typically encountered during rhododendron cultivation in container fields and greenhouses. Predation rates were assessed for third-instar larvae and adults of both sexes. The performance of *M. angulatus* third instars was also compared with that of second- and third-instar larvae of the green lacewing *Chrysoperla carnea* (Stephens) sensu lato (s.l.), a standard biological control agent commonly used to manage aphid outbreaks.

## 2. Materials and Methods

### 2.1. Insect Cultures

*Illinoia lambersi* individuals were collected in 2022 from infested rhododendron plants at Viaverda research station (Destelbergen, Belgium) and identified based on DNA barcoding using the COI gene. The aphids were reared on *Rhododendron simsii* ‘Nordlicht’ plants, ranging from bud stage to flowering, under controlled conditions: 23 ± 1 °C, 65 ± 5% relative humidity (RH), and a 16:8 h (L:D) photoperiod. Fertilization was applied weekly with liquid fertilizer for ericaceous plants from DCM (Grobbendonk, Belgium), except during flowering.

Larvae of *M. angulatus* were obtained from Biobest Group NV (Westerlo, Belgium). All life stages were kept in separate breeding containers (20 × 15 × 5 cm) with meshed lids (10 cm diameter), under controlled conditions of 25 ± 1 °C, 60 ± 5% RH, and a 16:8 h (L:D) photoperiod. Both adults and larvae were fed ad libitum with *Myzus persicae* (Sulzer) reared on broad bean (*Vicia faba*) plants. Two water sources, consisting of pieces of cotton soaked in water and placed in bottlecaps, were included in each container. Water and food were replenished three times a week. Shredded paper was added to the larval rearing containers to provide pupation sites and hiding places. In the adult rearing containers, cotton was attached to the lids to serve as an oviposition substrate [15]. Three times a week, the cotton pads with deposited eggs were transferred to new containers to allow hatching, and fresh cotton pads were provided.

To initiate a culture of *C. carnea* s.l., individuals were obtained from Biobest Group NV (Westerlo, Belgium; Chrysopa-System) and reared following the protocol of Berteloot et al. [19]. Adult lacewings were fed with commercial bee pollen grains (Weyn’s Honing, Beveren, Belgium), whereas larvae were provided with thawed *Ephestia kuehniella* Zeller eggs supplied by Koppert BV (Berkel en Rodenrijs, The Netherlands). The lacewings were kept in climatic chambers set at 25 ± 1 °C, 60 ± 5% RH, and a 16:8 h (L:D) photoperiod.

### 2.2. Predation Experiments

The predation efficacy of *M. angulatus* and *C. carnea* on *I. lambersi* was evaluated in climatic chambers maintained at 60 ± 5% RH, and a 16 h photoperiod, with temperatures varying according to the experiment. To assess the voracity of the lacewings and their potential applicability under various temperature conditions, laboratory experiments were conducted at temperatures ranging from 15 to 25 °C, using predation arenas.

Each arena consisted of an insect breeding dish (100 × 40 mm; SPL Life Sciences Co., Pocheon-si, Republic of Korea) in which two lateral holes (~0.75 cm in diameter) were created to facilitate the introduction of prey and predators without risk of escape. A fresh *R. simsii* ‘Nordlicht’ leaf, placed in an Eppendorf tube (0.6 mL) filled with water and stoppered with cotton, served as a food source for the aphids. In a preliminary trial, three initial densities (40, 60, and 80 aphids, always consisting of a mix of third instars, fourth instars, and wingless adults) were tested with a single predator in a dish arena. Sixty aphids were used in subsequent experiments, as 40 approached the maximum daily predation rate of a single larval or adult lacewing and 80 largely exceeded it. Hence, in the final experiments, 60 individuals of *I. lambersi* were added to each arena using a fine brush. Depending on the experiment, a single predator, either *M. angulatus* or *C. carnea*, of the various tested life stages as described below, was introduced. Ten replicates were conducted for each treatment, along with control treatments in which 60 aphids were placed in the arenas without predators.

To standardize hunger levels and allow predators to acclimatize to the test temperatures, all individuals were starved for 24 h at the temperature at which the predation experiment would take place. During this starvation period, the lacewings were isolated and had access to water, supplied via moist cotton fitted in bottlecaps. The arenas were then left undisturbed for 24 h, after which the predation efficacy of the tested life stage was assessed by counting the remaining aphids.

#### 2.2.1. Predation by *M. angulatus* and *C. carnea* Larvae

The predation efficacy of third-instar larvae (L3) of *M. angulatus* on *I. lambersi* was evaluated at three constant temperatures: 15, 20, and 25 ± 1 °C. For *C. carnea*, the predation efficacy of both second- (L2) and third-instar larvae (L3) was evaluated at two temperatures: 15 and 20 ± 1 °C. Third instars of both predators were selected, as they account for approximately 80% of total aphid predation in the larval stage [11,20]. Third instars of *M. angulatus* (with head capsule width of the tested individuals averaging 0.59 mm) are smaller than those of *C. carnea* (head capsule width of ca. 0.90 mm) but are comparable in size to second instars of *C. carnea* (head capsule width of ca. 0.60 mm); therefore, second instars of *C. carnea* were also included in the experiment. The temperatures were chosen to reflect cooler spring conditions (around 15 °C), moderate summer conditions (around 20 °C), and warmer summer conditions (around 25 °C) in northwestern Europe, in order to assess the potential effectiveness of these predators in the field under varying seasonal conditions. As we were mainly interested in comparing the predatory performance of *M. angulatus* larvae with those of *C. carnea* at low to moderate temperatures (late spring-early summer), larvae of *C. carnea* were not tested at 25 °C. Newly molted larvae (<24 h old) were taken from the communal rearings and subjected to a 24 h starvation period, during which they had access to water, before being tested as described above.

#### 2.2.2. Predation by *M. angulatus* Adults

The effect of sex and temperature on predation performance of *M. angulatus* adults was assessed with individuals from the stock colony that were 7–8 days post-emergence (i.e., reproductively active). To evaluate the effect of sex, male and female adults were tested at 25 ± 1 °C. Sex differentiation was conducted microscopically by checking for the presence of the arcessus, a hook-like copulatory organ in the males [21]. To examine the effect of temperature, female adults were subjected to 15, 20, and 25 °C ± 1 °C, as above; only female adults were selected for this experiment based on their higher predatory capacity, as evidenced by the first trial with adults. All adults underwent a 24 h starvation period before testing, and their predation rate was measured after another 24 h following the same procedure described above.

### 2.3. Statistical Analysis

All statistical analyses were conducted using R version 4.3.0 [22]. A significance level of 5% was applied for all experiments. Predatory efficacy was calculated for all predator treatments using Abbott’s formula [23] to account for natural mortality:*Corrected aphid mortality* (*in* %) = [(*M_treatment_* − *M_control_*)/(100 − *M_control_*)] × 100
where *M_treatment_* is the aphid mortality (*in* %) in the predator treatments and *M_control_* is the mean aphid mortality (*in* %) of the control treatments. Corrected aphid mortality was then compared between different treatments using a linear model with a normal distribution incorporating predator, temperature, and their interaction as fixed factors. To identify significant differences between specific treatments, pairwise comparisons were conducted using the marginal effects package.

## 3. Results

Control treatments (only *I. lambersi* aphids) for larval predation assays resulted in aphid mortality of 3.0 ± 0.8% (mean ± SE), 1.8 ± 0.8%, and 6.0 ± 1.8% at 15, 20, and 25 °C, respectively. Predation by third instars (L3) of *M. angulatus* over a 24 h period resulted in corrected *I. lambersi* mortality rates of 61.3 ± 10.5%, 51.4 ± 5.0%, and 87.2 ± 3.0% at the respective temperatures (Figure 1). For *C. carnea*, second-instar larvae (L2) caused corrected aphid mortality rates of 26.6 ± 4.2% at 15 °C and 26.3 ± 5.4% at 20 °C, while third instars caused higher mortality, amounting up to 57.9 ± 5.8% of the aphids offered at 15 °C and 72.8 ± 7.0% at 20 °C (Figure 1).

Control groups (aphids only) for adult predation experiments showed mortality rates of 5.2 ± 1.5%, 5.3 ± 1.4%, and 10.0 ± 2.7% at 15, 20, and 25 °C, respectively. Predation by adult females of *M. angulatus* led to corrected aphid mortality rates of 48.5 ± 5.4%, 50.5 ± 3.2%, and 63.9 ± 4.3% at the respective temperatures. Male adults, which were only tested at 25 °C, killed 38.9 ± 4.1% of the aphids offered (Figure 2).

A linear regression model showed a significant effect of predator type, temperature, and their interaction on aphid mortality (F(10, 99) = 10.82, *p* < 0.001). Pairwise comparisons revealed significant differences across temperature regimes and predator types.

For *M. angulatus* L3, aphid mortality increased significantly between 15 and 25 °C (*p* = 0.001) and between 20 and 25 °C (*p* < 0.001), while no significant difference was found between 15 and 20 °C (*p* = 0.213). In contrast, temperature did not significantly affect the predatory efficacy of adult females of *M. angulatus*. Although aphid mortality was higher at 25 °C compared to 15 °C, this difference was only marginally significant (*p* = 0.053). The differences in mortality between 15 and 20 °C (*p* = 0.799) and between 20 and 25 °C (*p* = 0.093) were not statistically significant. For both *C. carnea* L2 and L3 larvae, temperature (15 vs. 20 °C) had no significant effect on aphid mortality (*p* = 0.968 for L2, *p* = 0.060 for L3).

As to comparisons among predator stages and species, no significant difference in predatory efficacy on *I. lambersi* was observed between *M. angulatus* L3 and *C. carnea* L3 at 15 °C (*p* = 0.665), but both were significantly more effective than *C. carnea* L2 (*p* < 0.001 for both comparisons). At 20 °C, however, *C. carnea* L3 caused significantly higher aphid mortality than *M. angulatus* L3 (*p* = 0.007). Like at 15 °C, both *C. carnea* L3 (*p* < 0.001) and *M. angulatus* L3 (*p* = 0.002) outperformed *C. carnea* L2. Performance of female adults of *M. angulatus* was comparable to that of conspecific L3 larvae at 15 °C (*p* = 0.106) and 20 °C (*p* = 0.908). However, at 25 °C, L3 larvae of the brown lacewing were significantly more effective than female adults (*p* = 0.003). Male adults of *M. angulatus* at 25 °C were less efficient predators, showing significantly lower predatory efficacy compared to both L3 larvae (*p* < 0.001) and female adults (*p* = 0.002).

## 4. Discussion

Although commercial augmentative releases of *C. carnea* and other green lacewing species are an established component of integrated pest management programs targeting aphids and other arthropod pests worldwide [9], the brown lacewing *M. angulatus* has only recently been introduced to the biological control market [24]. This species is currently promoted as a generalist predator of aphids across a range of ornamental, fruit, and vegetable crops, both in the open field and in protected cultivation [25]. The present paper focused on the potential of *M. angulatus* to suppress populations of *I. lambersi*, a key aphid pest of rhododendron, and compared its performance at moderate to low temperatures with that of *C. carnea* s.l. Our laboratory experiments assessed the predation rates of third instars of *M. angulatus* and second and third instars of *C. carnea*, as well as of *M. angulatus* adults, on a mix of late instars and adults of *I. lambersi.*

Overall, the findings from our predation experiments are in line with [15,26], which reported that temperatures between 15 and 25 °C are suitable for the development and reproduction of *M. angulatus*. At the highest tested temperature (25 °C), third instars of *M. angulatus* consumed more aphids (ca. 52 late instar and adult aphids per day) than at cooler temperatures (30–37 aphids per day at 15 °C and 20 °C). On the other hand, temperature did not significantly affect predation by female adults. Female *M. angulatus* exhibited consistent predation rates across all tested temperatures, consuming approximately 30 to 40 aphids over a 24 h period (Figure 2). Despite their smaller size based on head capsule widths, third instars of *M. angulatus* exhibited similar predatory efficacy at 15 °C as third instars of *C. carnea*, with ca. 35 aphids killed per day. At 20 °C, however, *C. carnea* third instars showed somewhat higher predation rates on *I. lambersi* than those of *M. angulatus*, consuming around 43 vs. 30 aphids per day. Third instars of *M. angulatus* consistently outperformed similar-sized *C. carnea* second instars in terms of predation rates. Further, the third (final) instar of *C. carnea* was more voracious as a predator of rhododendron aphids at both 15 and 20 °C than the second instar, with two to three times higher predation rates. Male adults of *M. angulatus* demonstrated significantly lower predation rates than females, consuming only 23–24 aphids at 25 °C. Similar sex-based differences in predation have been observed for *Micromus timidus* Hagen, with female adults consuming 2–3 times more *Aphis gossypii* Glover or *M. persicae* aphids than males [27]. Higher predation rates in female adults of predatory insects compared with males are commonly attributed to the higher energetic demands of females associated with reproductive investment [28]. Finally, when interpreting the presented predation rates, it should be noted that the starvation period to which larvae and adults were subjected prior to the trials may have increased their short-term voracity relative to unstressed daily feeding rates.

Our findings corroborate those of an earlier semi-field study, indicating that aphid predation during larval development of *M. angulatus* was comparable to that of *C. carnea* in sweet pepper and strawberry [16]. Unlike *C. carnea*, however, which is only predaceous in the larval stage, *M. angulatus* attacks prey both in the larval and the adult stages. With an average larval period of ca. 7 days at 24 °C and an adult lifespan extending over 10 weeks [15,26], *M. angulatus* can contribute longer to pest suppression as compared with *C. carnea*. Whereas in the studies of [11,16] releases of *M. angulatus* as eggs, first instars and adults proved successful in suppressing aphids on sweet pepper and strawberry plants, the predator is presently commercially distributed as adults [25], with an approximate sex ratio of 1:1.

In northwestern Europe, pot azaleas and other rhododendrons are typically overwintered under protective cover and transferred to outdoor container fields in spring [1]. At this stage, they become susceptible to infestation by the first generation of winged *I. lambersi* adults (April-May in the Flanders region). Any natural enemies applied augmentatively against these aphids must be able to perform reliably under early spring conditions. In our experiments, third-instar and adult *M. angulatus*, as well as third-instar *C. carnea*, demonstrated effective predation on rhododendron aphids at a relatively low temperature of 15 °C. These findings indicate the potential of these predators to suppress aphid outbreaks early in the growing season. However, further validation under open-field conditions is required to confirm the results of our laboratory study.

Finally, our laboratory assays demonstrated that larvae of *M. angulatus* and *C. carnea* and adults of *M. angulatus* successfully preyed upon *I. lambersi* when the aphids were offered on leaves of *R. simsii* ‘Nordlicht’. This cultivar, characterized by relatively smooth leaf surfaces, provided minimal physical barriers to predator movement and feeding. However, the applicability of these findings to other rhododendron cultivars requires careful consideration. Many commercially important cultivars possess dense glandular trichomes on their leaves, which may impede predator locomotion, reduce prey encounter rates, or alter foraging efficiency [29]. In our preliminary laboratory trials, we observed that third-instar *C. carnea* frequently became immobilized in the sticky exudates present on the leaves of *Rhododendron ponticum* ‘Graziella’. Consequently, while our results confirm the potential of green and brown lacewings as biological control agents on smooth-leaved cultivars of *Rhododendron*, further research is needed to assess their performance on trichome-rich varieties under both controlled and field conditions.

## Figures and Tables

**Figure 1 insects-17-00046-f001:**
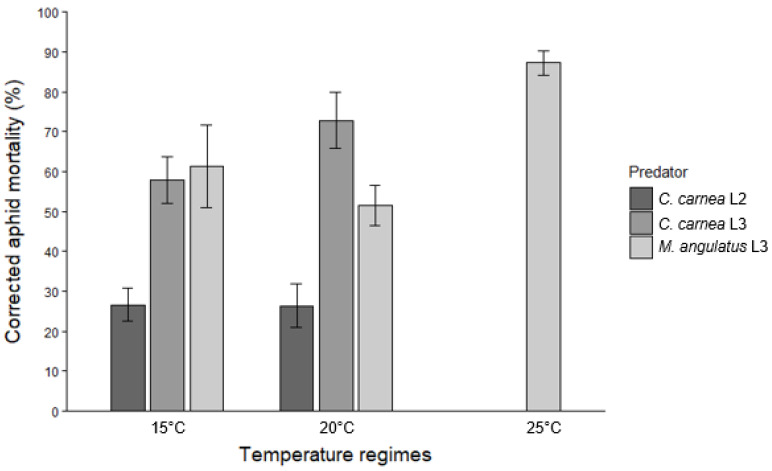
Corrected mortality of *I. lambersi* (means ± SE) after 24 h of predation by second- (L2) and third-instar (L3) larvae of *C. carnea* and third-instar larvae of *M. angulatus* at 15, 20, and 25 °C. Each predator was offered 60 aphids (a mix of third and fourth instars and wingless adults) initially.

**Figure 2 insects-17-00046-f002:**
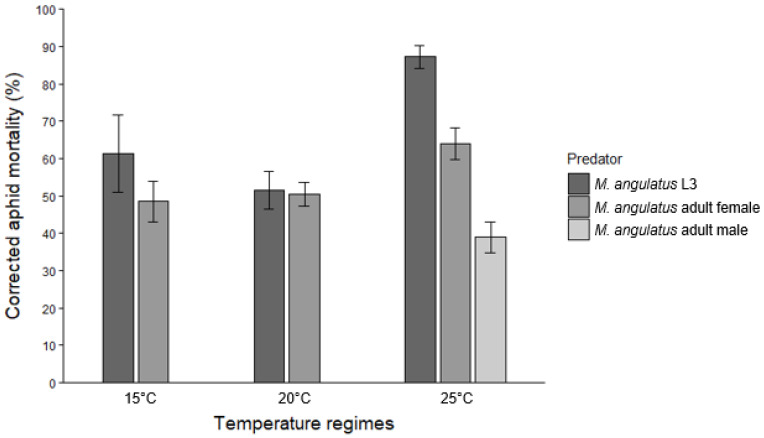
Corrected percentage of *I. lambersi* mortality (means ± SE) after 24 h of predation by third-instar larvae (L3) and female adults of *M. angulatus* at 15, 20, and 25 °C, and by male adults of *M. angulatus* at 25 °C. Each predator was offered 60 aphids (a mix of third and fourth instars and wingless adults) initially.

## Data Availability

The raw data supporting the conclusions of this article will be made available by the authors on request.

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
