# Peer review of "The Lacewings Micromus angulatus and Chrysoperla carnea as Predators of the Rhododendron Aphid, Illinoia lambersi, Under Different Temperature Regimes"

_insects, 2025, doi:10.3390/insects17010046_

Round 1
Reviewer 1 Report
Comments and Suggestions for Authors
The authors in paper entitled "The lacewings Micromus angulatus and Chrysoperla carnea as
predators of the rhododendron aphid, Illinoia lambersi, under different temperature regimes" presented findings in a clear and simple manner with adequate scientific soundness. Although trial was relatively complex (two different predators, different insect stages, two to three temperature regimes etc.) methodology is well explained. Results obtained via set of trials are leading to a proper conclusions. Literature cited is adequate.
Comments:
- In keywords terms from the title should be avoided.
- In material and methods, although not of great interest since that were laboratory trials, the year when trials were conducted could be mentioned.
- It would ensure even more compehensive results if C. carnea was tested under regime of 25C and more detailed comparison of two predators.
- In lines 166 to 168, the last sentence of the paragraph, are mentioned newly emerged larvae. Since trials were done with L2 and L3 why this is mentioned.
- In line 198 the authors should try to give an explanation why I. lambersi mortality was the lowest at the "medium" temperature of 20C (at 15 and 25C mortality is higher).
- Line 237 "Male adults of M. angulatus were less efficient". The authors should try to give the explanation why are males less efiicient. It is mentioned in discussion that another species from the same genus acts the same but without scientific explanation.
- Lines from 246 to 251 (last two sentences from the paragraph) are more appropriate for the Material and method part and should be ommited.
- Although the results are presented in a form of graphics all the results of predation summarised in one additional table could be a good option.
- The authors emphasised the importance of similar testing in open field conditions and on other Rhododendron varieties which is very proffesional and shows scientific awareness.
Author Response
We appreciate the constructive comments of the reviewer and have taken them into careful consideration when revising the manuscript.
Comment 1: In keywords terms from the title should be avoided.
Response 1: Done
Comment 2: In material and methods, although not of great interest since that were laboratory trials, the year when trials were conducted could be mentioned.
Response 2: We have provided a date for the start of our rhododendron aphid colony, which indicates the start of the experiments.
Comment 3: It would ensure even more compehensive results if C. carnea was tested under regime of 25 °C and more detailed comparison of two predators.
Response 3: We acknowledge that our experimental design was incomplete. We would have preferred to include a comparison of the larval performance of both lacewings at 25°C, but due to logistical and time constraints, gave priority to comparing predation rates of both species at the lower temperatures as this is more relevant for practice. We have explained this in the Materials and Methods section.
Comment 4: In lines 166 to 168, the last sentence of the paragraph, are mentioned newly emerged larvae. Since trials were done with L2 and L3 why this is mentioned.
Response 4: We have replaced "newly emerged" with "newly molted" to avoid any confusion.
Comment 5: In line 198 the authors should try to give an explanation why I. lambersi mortality was the lowest at the "medium" temperature of 20C (at 15 and 25C mortality is higher).
Response 5: The inference of the reviewer is not entirely correct, based on our statistical analysis. As is explained lower in the text, for third instar M. angulatus, aphid mortality was significantly different between 15 and 25 °C (p = 0.001) and between 20 and 25 °C (p < 0.001), while no significant difference was found between 15 and 20 °C (p = 0.213). So, whereas prey mortality at the medium temperature may have been the lowest in absolute numbers, it was not statistically different from that at the lower temperature and arguably we have to respect the outcome of the statistical analysis. Effects of temperature on predation rates are discussed along the same lines in the Discussion section.
Comment 6: Line 237 "Male adults of M. angulatus were less efficient". The authors should try to give the explanation why are males less efiicient. It is mentioned in discussion that another species from the same genus acts the same but without scientific explanation.
Response 6: We acknowledge that sexual differences in consumption rates can be biologically meaningful. In many predatory arthropods, females exhibit higher prey consumption than males under equivalent prey densities. This pattern is widely attributed to the greater energetic demands of egg production and reproductive investment in females, which require sustained nutrient intake, whereas male adults typically invest more energy in mate searching and less in sustained feeding. We have now inserted a sentence explaining this and have provided an extra reference.
Comment 7: Lines from 246 to 251 (last two sentences from the paragraph) are more appropriate for the Material and method part and should be ommited.
Response 7: While detailed methodological information is indeed provided in the Materials and Methods section, briefly summarizing the experimental framework at the start of the Discussion helps set the stage for interpretation and is not uncommon in experimental studies. We therefore prefer to retain this short contextual statement.
Comment 8: Although the results are presented in a form of graphics all the results of predation summarised in one additional table could be a good option.
Response 8: We appreciate the reviewer’s interest in the numerical predation rates. In our manuscript we present figures with prey mortality as percentages, which were used for all statistical analyses, and with the fixed initial prey density of 60 aphids readers can readily convert these to absolute numbers. In the second paragraph of the Discussion, we have already included the key predation values in the text to support interpretation by the reader. According to standard scientific writing guidance, redundant presentation of the same data in multiple formats (e.g., the same data both in a figure and in a separate table) is discouraged because it does not add new information. The instructions for authors of MDPI journals like Insects specifies that "images should not be duplicated". Hence, we understand also from the journal guidelines that an additional table containing the same predation numbers would constitute unnecessary duplication and therefore believe it is best not to provide this extra table.
Comment 9: The authors emphasised the importance of similar testing in open field conditions and on other Rhododendron varieties which is very proffesional and shows scientific awareness.
Response 9: We thank the reviewer for expressing his appreciation of our concluding remarks.
Reviewer 2 Report
Comments and Suggestions for Authors
This is a small, simple study evaluating the predation capacity of a less common biocontrol agent insect. The experimental design is not complete, but the absence of the missing treatments is explained. A linear model is generally not the best method for percentages, but with values around 50 % it can be used. Some small additions to the methods are needed, such as the preliminary results leading to the use of a single prey density of 60 individuals, while the feeding rate is known to change with prey density according to the Holling functional response curves. There were not so many individuals to allow for two decimal precision. Reduce the number of decimals. In discussion, authors must mention that the level of voracity in this design is higher than would be simple daily voracity without starvation.

Author Response
We appreciate the constructive comments of the reviewer and have revised the text accordingly.
Comment 1: The experimental design is not complete, but the absence of the missing treatments is explained.
Response 1: Indeed, we would have preferred to also perform experiments with C. carnea larvae at 25°C, but could not do this due to time constraints and thus gave priority to comparing the larval performance of both lacewings at the lower temperatures as this was more relevant from a practical viewpoint. We thank the reviewer for their understanding.
Comment 2: A linear model is generally not the best method for percentages, but with values around 50 % it can be used.
Response 2: The choice for this linear model was the result of several discussions we had on the data structure with our statistical consultant at Ghent University, Dr Dries Reynders. He also believed that the model we used would be appropriate for our percentage data, with most Abbott corrected mortality rates indeed ranging from 30 to 70%.
Comment 3: Some small additions to the methods are needed, such as the preliminary results leading to the use of a single prey density of 60 individuals, while the feeding rate is known to change with prey density according to the Holling functional response curves.
Response 3: We acknowledge that prey density can influence predator functional responses. The purpose of the preliminary trial was therefore not to characterize functional responses across densities, but to select a prey density that would ensure ad libitum conditions throughout the experiment while avoiding prey limitation. Three initial prey densities (40, 60, and 80 aphids) were tested in Petri dish arenas with a single predator. At 40 aphids, prey availability was close to the predator’s maximum daily predation rate, increasing the risk of prey depletion and confounding treatment effects through prey limitation. In contrast, 80 aphids greatly exceeded the predator’s daily consumption capacity and would have required substantially increased prey rearing without additional experimental benefit. A density of 60 aphids was therefore selected as a conservative compromise, ensuring continuous prey availability while minimizing logistical constraints. Under these conditions, prey density was not intended to be a limiting factor, thereby reducing its influence on observed predation rates. In compliance with the reviewer's request, we have now added a description of this preliminary experiment (lines 137-141).
Comment 4: There were not so many individuals to allow for two decimal precision. Reduce the number of decimals.
Response 4: We agree and have now reduced the number of decimals to one.
Comment 5: In discussion, authors must mention that the level of voracity in this design is higher than would be simple daily voracity without starvation.
Response 5: Thank you for this comment. We have now inserted a sentence in the Discussion highlighting that voracity may indeed have been affected by the starvation treatment (lines 276-279).